# Association of *NUDT15* c.415C>T and *FPGS* 2572C>T Variants with the Risk of Early Hematologic Toxicity During 6-MP and Low-Dose Methotrexate-Based Maintenance Therapy in Indian Patients with Acute Lymphoblastic Leukemia

**DOI:** 10.3390/genes11060594

**Published:** 2020-05-28

**Authors:** Sunitha Kodidela, Patchava Dorababu, Dimpal N. Thakkar, Biswajit Dubashi, Rajan Sundaram, Niveditha Muralidharan, Ravi Prasad Nidanapu, Anil Aribandi, Suresh Chandra Pradhan, Chakradhara Rao Satyanarayana Uppugunduri

**Affiliations:** 1College of Pharmacy, University of Tennessee Heath Science Center, Memphis, TN 38163, USA; 2Department of Pharmacology, Apollo Institute of Medical Sciences and Research, Hyderabad 500090, India; patchvadora@gmail.com; 3Department of Pharmacology, Jawaharlal Institute of Postgraduate Medical Education & Research, Pondicherry 605006, India; dimpalnthakkar@gmail.com (D.N.T.); ss_rjn@yahoo.com (R.S.); ravipisces19@gmail.com (R.P.N.); 4Department of Medical Oncology, Jawaharlal Institute of Postgraduate Medical Education & Research, Pondicherry 605006, India; drbiswajitdm@gmail.com; 5Department of Clinical Immunology, Jawaharlal Institute of Postgraduate Medical Education & Research, Pondicherry 605006, India; bioniveditha@gmail.com; 6Division of Haemato-Oncology, Care Hospitals, Hyderabad 500019, India; aribandia@hotmail.com; 7American Oncology Institute, Nallagandla Serilingampalli, Hyderabad 500019, India; 8Department of Pharmacology, Kalinga Institute of Medical Sciences, Bhubaneswar 751024, India; Pradhanjipmer@gmail.com; 9Onco-Hematology Unit, Research Platform of Pediatric Onco-Hematology, Department of Paediatrics, Gynaecology and Obstetrics, University Hospitals of Geneva, University of Geneva, 1205 Geneva, Switzerland

**Keywords:** NUDT15, polymorphism, toxicity, 6-mercaptopurine, relapse, survival, methotrexate, myelosuppression, toxicity, acute lymphoblastic leukemia, children, India

## Abstract

Genetic variants influencing the pharmacokinetics and/or pharmacodynamics of the chemotherapeutic drugs used in Acute Lymphoblastic Leukemia (ALL) therapy often contribute to the occurrence of treatment related toxicity (TRT). In this study, we explored the association of candidate genetic variants with early hematological TRT (grade 3–4) occurring within the first 100 days of low-dose methotrexate and 6-mercaptopurine based maintenance therapy (*n* = 73). Fourteen variants in the following candidate genes were genotyped using allele discrimination assay by real-time PCR: *ABCB1*, *DHFR*, *GGH*, *FPGS*, *MTHFR*, *RFC1*, *SLCO1B1*, *TPMT*, and *NUDT15*. Methotrexate polyglutamate (MTXPG3-5) levels in red blood cells were measured by LC-MS/MS. Early hematological TRT (grade 3–4) was seen in 54.9% of patients. The *NUDT15*c.415T allele was associated with early TRT occurrence [HR: 3.04 (95% CI: 1.5–6.1); *p* = 0.007]. Sensitivity of early TRT prediction improved (from 30.7% to 89.7%) by considering *FPGS* variant (rs1544105’T’) carrier status along with *NUDT15*c.415T allele [HR = 2.7 (1.5–4.7, *p* = 0.008)]. None of the considered genetic variants were associated with MTXPG3-5 levels, which in turn were not associated with early TRT. *NUDT15*c.415T allele carrier status could be used as a stratifying marker for Indian ALL patients to distinguish patients at high or low risk of developing early hematological TRT.

## 1. Introduction

Globally, the survival of patients with acute lymphoblastic leukemia (ALL) has increased tremendously with the five-year overall survival (OS) reaching 90% in high-income countries [1,2]. However, in developing countries like India, the five-year OS has been reported to be 30–70% [3,4]. Moreover, deaths due to treatment-related toxicity (TRT) in Indian ALL patients ranges between 2%–24% [5], which is ten times higher than in high-income countries [6]. Multicentre protocol (MCP 841) for pediatric population and German multicenter ALL (GMALL) protocol for adults were developed as effective tools to treat ALL patients in the absence of risk stratification criteria, and for a cost-effective strategy in limited-resource settings [7]. These two protocols, include the administration of drugs that require minimal supportive care therapy, thus reducing the costs of treatment. Until recently, MCP-841 was used to treat the majority of childhood ALL patients in India. Event-free survival with the MCP-841 protocol ranged from 41% to 70% (average > 50%), whereas toxic deaths (i.e., non-relapse mortality) ranged from 2% to 13% during induction, and from 4% to 24% during ongoing treatment. Relapse rates with the MCP-841 protocol ranged from 18% to 41% (average 30%) [8,9]. Maintenance therapy in MCP-841 and GMALL protocols (Appendix A) consists of the daily oral dose of the low-intensity cytotoxic agent 6-mercaptopurine (6-MP), which belongs to the drug class thiopurines, and a weekly oral dose of the folate analog methotrexate (MTX). A combination of 6-MP and MTX is recommended, due to their pharmacokinetic (PK) and pharmacodynamic (PD) advantages. MTX inhibits xanthine oxidase and thereby increases 6-MP bioavailability, while also contributing to decreased 6-MP metabolite methylation by lowering the availability of S-adenosyl methionine (Appendix A) [10,11,12,13,14,15,16]. 

Despite the use of less intensive protocols in India compared to those used in high income countries, there are higher incidences of death due to toxicities, especially life-threatening myelosuppression and infections (17–22%) [9,17,18]. However, the incidence of myelosuppression is low with the use of low-dose MTX (LDMTX). Altered levels of MTX polyglutamates, along with altered PK or PD of 6-MP, can predispose some patients to develop myelosuppression. Identifying such high-risk patients will be clinically significant to take appropriate preventive measures, such as the administration of 6-MP and MTX in optimal doses from the start of maintenance therapy to reduce the incidence of myelosuppression in Indian ALL patients. Patients respond differently to the same protocol, hence, it is imperative to elucidate the factors contributing to the observed variability in the safety and efficacy of the 6-MP and MTX in high- and low-risk populations to achieve optimal outcomes. One of the factors involved could be genetic variations in the proteins, which are responsible for determining the respective PK and PD of those two drugs. The cellular and metabolic pathways of 6-MP and MTX and the interactions between these drugs are detailed in previous publications, as depicted in Appendix A [10,15,19,20]. 

Recent reports indicate the importance of highly prevalent variants in nucleoside diphosphate-linked moiety X-type motif 15 (*NUDT15*) among Asians and Indians to predict 6-MP-induced myelosuppression [21,22,23]. This is in contrast to the Caucasians, where common thiopurine methyltransferase (*TPMT*) variants were shown to be associated with 6-MP-induced myelosuppression [24]. Therefore, we selected *NUDT15* (rs116855232) and *TPMT* (rs1142345) genetic variants, which are reported affect 6-MP outcome [25,26] (Table 1). Several variants in genes encoding metabolism, transporter proteins and in the folate pathway (Table 1) may play a role in determining toxicity and efficacy of the MTX. Therefore, we selected the common variants in genes involved in DNA synthesis and methylation (*DHFR* and *MTHFR*; Table 1), MTX polyglutamation (*FPGS* and *GGH*; Table 1), and MTX transporters (*RFC1*, *SLCO1B1*, and MDR1 or *ABCB1*; Table 1). Evaluating the role of these common genetic variants of the vital proteins and determining the PK and PD of 6-MP and MTX (Table 1) is essential in understanding the drug-drug and gene-gene interactions, which will allow for prediction and prevention of early TRT. To the best of our knowledge, no such studies have been conducted in South Indian ALL patients. Hence, this explorative pilot study was conducted to evaluate the association of fourteen candidate variants with early hematological TRT. The association with other clinical outcomes, such as incidence of relapse was also explored. 

## 2. Materials and Methods 

### 2.1. Patients and Inclusion Criteria

Institutional scientific and ethics committees approved this observational cohort study (JIP/IEC/SC/2/2012/28). A total of 73 subjects with ALL were enrolled in the study between May 2012 and January 2016, after obtaining written informed consent from the patients or, in the case of children, from legally accepted guardians. Patients who achieved complete remission after the induction and consolidation phases of chemotherapy receiving 6-MP- and LDMTX-based maintenance therapy were included in the study. All subjects were of South Indian origin, with at least three generations living in the southern part of India, speaking the native language of the region. Patients with renal and hepatic dysfunctions, patients who were younger than one year of age, female patients who were pregnant or lactating, and patients who had any other active malignancies were excluded from the study.

### 2.2. Sample Collection and Genotyping of Candidate Variants

A total of 10 mL of EDTA-anticoagulant blood was collected from each patient once they had undergone complete remission. A 5 mL sample of whole blood was used for genotyping of the selected variants from genomic DNA. Another 5 mL of blood was collected after achieving MTX steady-state concentration (6–12 weeks) [27,28] to measure methotrexate polyglutamate (MTXPG) levels. Genomic DNA was extracted from whole blood by the phenol-chloroform method. The genotyping of selected genetic variants (Table 1) was carried out using validated TaqMan^®^ allele discrimination assays according to the manufacturer instructions (Applied Biosystems; Foster City, CA, USA) on a real-time PCR instrument (ABI Prism 7300; Foster City, CA, USA). Positive and negative controls were included in all genotyping assays. The quantification of MTXPGs3-5 in red blood cells (RBCs) was determined using high-performance liquid chromatography coupled to electrospray-ionization tandem mass spectrometry (Waters; Milford, MA, USA), as described previously [29] with minor modifications. Intra-and inter-day coefficients of variation were less than 15%. The results were dose-normalized, and these values were normalized to RBC count so that the results were comparable and not confounded by differences in dose or RBC count between individuals. 

### 2.3. Treatment Protocol and Clinical Outcomes

Patients below 25 years of age were treated using MCP-841 and those older than 25 years were treated with modified GMALL-84 protocols (Appendix A). LDMTX and 6-MP doses did not differ significantly between the two protocols during maintenance therapy. The primary clinical outcome assessed was incidence of early TRT (grade 3–4). Toxicities were graded according to the Common Terminology Criteria for Adverse Events (CTCAE-version 4.2). The toxicity, arising from its genetic predisposition, was hypothesized to be observed during the early stages of treatment, when the influence of patient and concomitant treatment-related factors is most likely to be minimal. Therefore, grade 3–4 hematological toxicities, occurring within the first 100 days of the maintenance therapy, were considered to be early TRT. We also assessed hepatic and renal toxicity in this cohort. The incidence of relapse was also explored in this cohort as a secondary clinical outcome. Relapse is defined as the recurrence of disease after complete remission, meeting at least one of the following criteria: (a) ≥ 5% blast presence in the marrow or peripheral blood, (b) extra-medullary disease, or (c) disease presence determined by a physician upon clinical assessment. Cumulative incidence of relapse was calculated from the start of maintenance therapy until the day of occurrence of relapse. Overall survival and relapse free survival are measured at five years from the start of the maintenance treatment. Other outcomes, such as hepatic toxicities, were also recorded as per CTCAE-version4.2. 

### 2.4. Statistical Analyses

The observed genotype frequencies were tested for Hardy–Weinberg Equilibrium (HWE) using the chi-square test. The association of genotypes and clinical factors, with early TRT, was explored using Fisher’s exact test (categorical) or the Mann-Whitney U-test (continuous variables). We explored the distribution of Fisher’s exact test *p* values for their association with early TRT among genotype groups. As expected, many hypotheses failed, thereby potentially representing false negatives due to low sample size. Cumulative incidences of early TRT and relapse among groups were obtained using the cumulative incidence function in competing risk package (cmprsk) in R [30,31]. Following the Benjimani-Hoechberg procedure, false discovery rate correction [32] for multiple comparisons was performed and *p* values corrected for multiple testing were obtained. The statistical significance level was set (multiple testing corrected) at <0.05. Risk factor analysis and hazard ratios were calculated using Cox regression. The Mann-Whitney U test (two groups) and Kruskal-Wallis tests (more than two groups) were used to compare drug levels with toxicity and relapse, and to compare drug levels across various genotype groups. A receiver–operator characteristic curve (ROC) for genetic markers, was plotted to show the trade-off in sensitivity versus 1- specificity rates for early TRT, for analyzing gene-gene interaction comparisons. Individuals with non-missing genotypes (*n* = 71) were included for genetic association analyses. Data analyses were carried out using the statistical software “R” version 2.13.1 and SPSS software (Version 26). Since this is a pilot explorative study, *a priori* calculation of power and sample size was not performed (please refer to the discussion section).

## 3. Results

A total of 73 patients with ALL with a mean age at diagnosis of 14.5 (SD, 11.4) years were included in this study. Among them, 18 were above 18 years of age, and 55 were below or equal to 18 years of age. The distributions of demographic and genotype variables between patients with severe early TRT and those with mild or no toxicity are given in the Table 2.

Grade 3–4 hematological toxicity was observed in 40 (54.9%) patients with the remainder having no or mild toxicity (grade 1–2) in the early phase of maintenance therapy. Early TRT cumulative incidence curve *p* values for each variant, and corrected *p* values after multiple testing, are given in Table 3. Among the studied genetic variants, *NUDT15*c.415T allele carriers had an increased risk for TRT (Table 2 and Table 3, Figure 1). 

The cumulative incidence of TRT was three fold higher in *NUDT15*c.415T allele carriers compared to non-carriers (Figure 1A). *FPGS* rs1544105 variant carriers (CT & TT genotypes) are at a higher risk of developing early TRT compared to patients with the normal genotype (CC) (Appendix A). Similarly, higher platelet count at the time of diagnosis was associated with reduced incidence of early TRT (Appendix A). After multiple testing correction, carrier status of *NUDT15* rs116855232(c.415C>T) variant and combined *NUDT15* rs116855232(c.415C>T) and *FPGS* rs1544105 loci remained significant (Table 3). The sensitivity to predict early TRT by *NUDT15* c.415T allele (Area under the curve = 0.623; 95% CI 0.538–0.707) was further improved when combined with that of the *FPGS* variant rs1544105‘T’ (Area under the curve of 0.691, 95% CI 0.59–0.792) in ROC curve analysis, but not statistically significant. Higher incidence of early TRT occurred in patients carrying variant alleles at either (44.44%) or both (90.90%) of the *NUDT15* rs116855232(c.415C>T) and *FPGS* rs1544105 loci, compared to normal allele carriers at both of these loci (26.66%; Figure 1B).

### 3.1. Association of Genetic Variants with 6-MP and MTX Dose Reduction within 100 Days of Maintenance Therapy

The dose details of 6-MP and MTX were available for 68 and 69 patients, respectively, during phase I of maintenance therapy (100 days). Out of those patients, 6-MP and MTX doses were reduced in 12, and 10 patients, respectively. As expected, there was an association between the occurrence of early TRT and 6-MP dose reduction (Table 2). The average dose of 6-MP within 100 days (I maintenance cycle) of maintenance therapy did not vary between *NUDT15*c.415T allele carriers (4529.8 mg/m^2^) and wild type allele carriers (4471.3 mg/m^2^). Similarly, there was no significant difference in dose reduction (*n* = 67; *p* = 0.1) between *NUDT15* c.415T allele carriers (7.6%) and non-carriers (19%). *FPGS* rs1544105 variant ‘T’ allele carriers (TT + TC) and normal ‘CC’ genotype carriers had a mean MTX dose of 211.8 mg/m^2^, and 169.2 mg/m^2^, respectively. Although the *FPGS* rs1544105 variant allele ‘T’ carriers had a higher percentage of dose reduction (13.4%) compared to normal genotype carriers (6.2%), and the difference was not statistically significant. Sixty-six patients were profiled for hepatic toxicity, but because of the low number of events (< 10), statistical analysis was not performed. None of our study participants experienced severe renal toxicity throughout the course of therapy. 

### 3.2. Association of Clinical and Genetic Variables with Relapse 

Of the 73 ALL patients the disease relapsed in 28 patients. The percentage distribution of age and sex did not differ between patients with or without relapse. The proportion of relapse and non-relapse cases that had blasts on day 8 was 89.7% and 86.4%, respectively. From the relapse cases (*n* = 28), 55.2%, and 31% were B, and T-ALL, respectively, with the patients’ immunophenotype information unavailable for the remaining 13.8%. Demographic and clinical characteristics did not differ significantly between relapse and non-relapse groups except for the white blood cell (WBC) count at the time of diagnosis and *ABCB1* is also known as *MDR1*c.3435 T>C variant genotype (Appendix A). Patients with WBC count > 50,000 cells/mm^3^ at the time of diagnosis had 3.6 times higher incidence of relapse compared to patients who had WBC counts of ≤50,000 cells/mm^3^ (Appendix A). Among the studied genetic variants, *ABCB1* 435 CCgenotype carriers were more likely to develop relapse compared to normal allele (CT & TT) carriers (Appendix A). The dose of 6-MP and MTX between patients with and without relapse was not significantly different. Further, the dose reduction of these drugs was not associated with the occurrence of relapse. These association analyses are explorative in nature, and hence, multiple testing correction was not applied. We also observed slightly higher incidence of relapse reflected as lower relapse free survival therapy in patients experiencing early TRT (55.7% vs. 66.5%) (Appendix A). Similarly, non-relapse mortality (4%) occurred only in patients who had early TRT; overall survival was lower in patents with early TRT compared to the rest (51.5% vs. 66.0%), however, these differences are not statistically significant (Appendix A).

### 3.3. Association of Methotrexate Polyglutamate Levels with ALL Relapse and Toxicity

MTXPG3-5 levels were measured in 55 patients after achieving steady-state concentration (6–8 weeks). Among all the MTXPGs, PG_3_ was the predominant metabolite, and PG_4_ and PG_5_ levels were undetectable in 29, and 37 patients, respectively. MTXPG3-5 levels were not associated with relapse (Appendix A). After three months of treatment, MTXPG3-5 levels were not found to be associated with hematological toxicity in our study (Appendix A). No significant differences were observed in the MTXPG3-5 levels among different genotype carriers of the studied MTX transporter or metabolizing gene variants (Appendix A). Similarly, MTXPG levels were not different between patients with severe and mild hepatic toxicity (data not shown). 

## 4. Discussion

In the present study, we showed that *NUDT15* rs116855232(c.415C>T) variant, combined with the *FPGS* rs1544105‘T’ variant allele could predict the occurrence of early TRT during maintenance treatment in patients with ALL. *TPMT**3C allele frequency was found to be low in this population (1.4%). A genome-wide association study recently demonstrated for the first time that a missense variant (rs116855232, c.415C>T) in the *NUDT15* gene is associated with 6-MP intolerance in patients with ALL [25]. Although genetic variations in *NUDT15* are rare, there are about 76 variants identified among which approximately 50% are deleterious to function due to reduced protein stability. However, the variants leading to deleterious function are more frequent in Asian populations, including Indians [21,33,34], compared to Caucasians and Africans [25]. The frequency of the *NUDT15*c.415T allele in the present study is 9.8%, and we did not observe any homozygous variant genotype carriers. This observed frequency is similar to another report from North India (9.5%) [21], and higher than that of the Middle Eastern population (0.7%) [26], but lower than those of the Chinese and Japanese populations (15–20%) [35]. These observations highlight the importance of *NUDT15* variants in predicting early TRT during maintenance therapy in ALL patients of South Indian origin, similar to the other Asian populations [21,33,34]. In the present study, the most common variant of *NUDT15*, the c.415C>T allele, has been studied. *NUDT15* rs116855232(c.415C>T) variant is present on both *2 and *3 haplotypes. However, we didn’t study the other locus (55_56insGAGTCG (V18_V19insGV) to differentiate *2 from *3 haplotype [36,37]. Further, the existence of other variants in this gene should not be overlooked, and the presence of such variants might explain TRT occurrence in individuals without c.415TC>T variant. One of the two individuals carrying the *TPMT**3C allele had TRT, but the other one did not. Thus, carrying any one of these alleles puts the patient at a higher risk of developing early TRT. This variant can serve as a stratifying marker as reported by our study and others [21,33,34,38]. Further, it is worth noting that all patients in our study received the protocols for minimum intensities with no risk stratification criteria during all phases of maintenance therapy (Appendix A).

NUDT15 inactivates thiopurine metabolites by converting thioguanosinetriphosphate to thioguanosinemonophosphate, thus, preventing their incorporation into DNA and leading to reduced cytotoxicity. Dysfunctional NUDT15 escalates the conversion of thioguanine (TGN) into DNA-TG, i.e. TGN incorporated into the DNA (Appendix A), thereby increasing the risk for development of toxicity [23]. This toxicity risk is even greater in patients with low TPMT activity and receiving combined MTX therapy, as compared to monotherapy of 6-MP. An erythrocyte 6-TGN concentration of <235 pmol/8 × 10^8^ RBCs has been reported to be associated with 6-MP treatment failure, whereas a 6-TGN concentration of >450 pmol/8 × 10^8^ RBCs is associated with a higher risk of leukopenia [39]. A wide range of toxicities have been observed across a range of 6-TGN levels, especially in patients receiving 6-MP for inflammatory bowel disease [40,41]. Interestingly, leukemia patients carrying dysfunctional *NUDT15* who experience leukopenia are reported to have lower 6-TGN levels (maximum of 171 pmol/8 × 10^8^ RBCs), which is far below the 450 pmol/8 × 10^8^ RBCs concentration threshold [23,34]. Such discrepancies indicate the importance of DNA-incorporated TGN levels, or the ratio of 6-TGN to DNA-TG, instead of free TGN levels in predicting therapeutic response and toxicity to 6-MP. Therefore, *NUDT15* genotyping offers benefits before administration of thiopurine or dose adjustments, regardless of free 6-TGN levels [23,34]. The Clinical Pharmacogenetics Implementation Consortium published peer-reviewed and evidence-based gene/drug clinical practice guidelines and recommended genotyping of *NUDT15* before 6-MP dosing [24].

We also observed that carriers of *FPGS* rs1544105 ‘T’ variant allele had an increased trend towards risk of early TRT. In a recent study conducted in patients with rheumatoid arthritis, it was shown that rs1544105 ‘CT’ genotype carriers of the *FPGS* gene, but not ‘TT’ carriers, had an increased risk of MTX-induced toxicity. The authors of that report included multiple types of toxicities, including hepatic, hematological, and gastrointestinal, without grading their severity [42]. In the present report, we considered only severe hematological toxicity as a phenotype for association in ALL patients receiving other combination medications that result in myelosupression, primarily 6-MP. The heterozygosity observed in our study (56%) is comparable to that reported by Muralidharan et al., [42]. The FPGS rs1544105 ‘T’ variant leads to decreased transcription and hence reduced function of FPGS. In the current report, we did not find a significant association between *FPGS* polymorphisms and MTXPG3-5 levels. FPGS converts MTX to MTXPGs and allows accumulation of intracellular MTX PGs that inhibit enzymes involved in the folate pathway. MTXPGs levels are not only regulated by the FPGS, and other proteins such as GGH, transporter proteins (MDR1, MRP2) might also contribute to the variability of intracellular levels of MTX. Due to the limited number of samples we did not analyze correlation of MTXPG levels in a gene–gene interaction model including all variants in *FPGS*, *GGH* and *MDR1*. Moreover, it had been reported that the ratio of MTXPGs 3–5 to MTXPGs1-2 could better predict the adverse effects related o MTX treatment in subjects carrying FPGS variant [43,44]. However, we didn’t measure MTXPG1-2 levels in our cohort. In the present study, combining *FPGS* rs1544105 ‘T’ allele with the *NUDT15* c.415T allele improved the sensitivity from 30.7% (*NUDT15**3) to 89.7% (G2; Carrying variant allele either in *FPGS* rs1544105 and *NUDT15* rs116855232(c.415C>T), Figure 1B) to predict the occurrence of early TRT. However, specificity is compromised from 93.8% (*NUDT15*c.415C>T) to 34.5% upon addition of *FPGS* rs1544105 ‘T’ variant along with *NUDT15c.415C>T* to predict TRT in ROC curve analysis (G2 in Figure 1B). This indicate role of other influencing factors in *FPGS* variant carriers who are not carrying *NUDT15*c.415C>T variant. Contradictory results exist regarding the association of *FPGS* rs1544105 polymorphisms with sensitivity and toxicity of MTX [22,45]. This could be due to differences in measured endpoints, in treatment regimens with different MTX doses, in allele frequencies, and in Linkage Disequilibrium (LD) pattern of SNPs between studied populations [46]. For e.g., variable LD patterns are seen between *FPGS* variants rs1054774 and rs1544105. These two loci are in LD among several populations but not in African populations [47]. Possible role of variants in other genes such as cyclin dependent kinase 9 (*CDK9*) that are in LD with that of *FPGS* rs1544105 variant on the same chromosome (data not shown) cannot be overlooked and might contribute to the observed variability in MTX toxicities and efficacy. However, functional evidence for the variants in *CDK9* intronic or regulatory region need to be investigated in future.

Early TRT seen during maintenance therapy of ALL cannot be attributed to a single therapeutic agent due to the synergistic nature of 6-MP and MTX. It is known that MTX increases the bioavailability of 6-MP by inhibiting xanthine oxidase (XO or XDH), consequently enhancing the risk of developing 6-MP-induced myelosuppression [10,11,12,13,14,15,16]. FPGS is predominantly expressed in gastro-intestinal tract (GIT), liver and lymphoid tissues compared to the other tissues, whereas XDH or XO is highly expressed in GIT [48]. The oral as well intravenous administration of MTX could inhibit XO in the gut [16], thus increasing the bio-availability of 6-MP. In addition to it, FPGS rs1544105 variant allele carriers perhaps have decreased expression of FPGS in the GIT which could probably lead to decreased conversion of MTX to MTXPGs, thus increasing the availability of MTX to inhibit the inactivation of 6-MP by XO. Further, depletion of S-adenosyl methionine by MTX also contributes to this observed interaction consequently (Appendix A). The observed TRT association with *FPGS* variants could plausibly be explained by this hypothesis and must be further investigated in a gene-gene interaction model, encompassing all possible partner genes in several pathways of these two drugs.

We did not observe an association between dose reduction and genetic variants evaluated in the study. Average 6-MP doses during maintenance therapy (100 days) were similar between *NUDT15*c.415T allele carriers and non-carriers, indicating that dose reduction in the carriers from the beginning may have prevented occurrence of early TRT. Clinical dose titration is challenging as it can sometimes lead to under- or overdosing, resulting in either relapse of disease or development of severe toxicities of 6-MP and MTX. 6-MP and MTX doses were reduced in 12 and 10 patients, respectively. The interruption of 6-MP and MTX maintenance therapy was primarily due to febrile neutropenia; other courses of interrupted therapy include lack of knowledge of caretakers about the dosing of the medicine, increased levels of liver enzymes (AST and ALT), and infections. Based on these observations, we hypothesize that early TRT during maintenance therapy, and associated complications, often discourage patients from continuing to receive treatment and/or to take medicine for the recommended timeframe (i.e., lack of compliance), resulting in disease progression, failure to follow-up, and possible death due to relapse or lack of supportive care. A trend has been demonstrated of higher incidences of relapse and of non-relapse mortality in patients who experienced early TRT in our study, although not statistically significant, which may support the idea that early TRT influencing OS and event-free survival. In this study, we observed lower OS in patients with early TRT (52.5%) compared to those without early TRT (66.7%), although these results were not statistically significant due to sample size. 

Identifying high-risk patients before initiation of therapy may aid in titrating the dose to avoid early toxicities. Apart from clinical risk factors, nutritional status and toxicities might lead to poor compliance and thus poor outcomes. Several nutritional deficiencies and body composition changes can affect the outcome of pediatric ALL treatment protocols [49]. Change in body size related to nutritional status is an essential factor governing drug disposition, as it alters the distribution and clearance [49]. For instance, subjects over- or underweight at the time of treatment initiation tend to have a higher risk of hepatic and pancreatic/pulmonary toxicities, respectively [50]. If such individuals also carry any one or more risk genetic variants, the odds for occurrence of toxicities in those patients would be enormously higher. Therefore, future studies must take this into consideration to understand and build gene-environment interaction models that will help to predict toxicities and improve outcomes. 

We also explored the association between clinical and genotype variables with relapse. We noted that the *ABCB1* 3435 variant genotype (CC) and a higher WBC count at the time of diagnosis are associated with an increased incidence of relapse. Though the 3435 T>C variant does not result in amino acid change, it may affect the activity of the MDR1 transporter by altering co-translational folding [51]. The *ABCB1* 3435 ‘TT’ genotype was found to reduce MDR expression and increase absorption of MDR1 substrates administered orally [52], which could contribute to better outcomes by possibly preventing the export of parent drugs and their metabolites from the cell. However, we did not observe any association between MTXPGs and *ABCB1* 3435 polymorphism. Similarly, other polymorphisms were also not associated with MTXPGs. Altered levels of MTXPGs, along with altered PK for 6-MP can predispose some patients to develop myelosuppression (Appendix A).

The known risk factors for ALL outcomes in the western population are age, sex, and WBC count, but these factors may vary across developing countries for various reasons [17]. Children aged 1–9 years tend to have a better chance of survival than patients who are younger than one year, or older than nine years. However, in the present study, age did not appear to affect the prognosis of the disease, similar to other studies conducted in India [17,53]. Females with ALL may have higher cure rates than males. However, we did not observe any difference in survival between females and males, which could be due to a smaller number of female subjects (38%) in the cohort relative to male subjects (62%). Moreover, as risk stratification criteria were not implemented in the treatment protocol, the observations of relapse incidence must be considered with caution and need to be replicated in a larger cohort of patients receiving risk stratification-based treatment protocols. 

The major limitation of the present pilot study is the small sample size. 6-MP and TGN levels were not measured. A posteriori calculation of the power with the observed 50% toxicity incidence (population risk); *NUDT15*c.415T had an allele frequency of 9.8% in the dominant mode of inheritance, for a genetic risk of 1.8 (lower limit of 95% CI), with a two-sided *p*-value of 0.05, which indicated that we need at least 204 samples to achieve 80% power. Power calculation for the present pilot explorative study sample size (*n* = 73) showed limited power that could explain non-associations of other known risk factors, such as age. However, the association found with *NUDT15* still remains, as it is evident within the sample of limited power. To address the limited power, and based on the observations from this pilot study, we are initiating an observational cohort study to evaluate the role of germline genetic variants in relation to treatment response and early toxicities in pediatric ALL patients in India. 

## 5. Conclusions

To conclude, *NUDT15* rs116855232(c.415C>T) and *FPGS* rs1544105(g.2572C>T) variants are promising targets for further evaluation. These two variants could serve as stratifying markers at the time of diagnosis to identify ALL patients who are at a higher risk to develop early hematological TRT, especially during 6-MP and LDMTX-based maintenance therapy. Implementation of preemptive *NUDT15* and *FPGS* genotyping to guide 6-MP dosing is worthy of prospective evaluation in limited resources settings, especially in terms of health economic evaluation. Therefore, pharmacogenetic information can offer considerable advantages in a limited-resource setting to identify the needs of individual patients, based on their genetic risk profiles.

## Figures and Tables

**Figure 1 genes-11-00594-f001:**
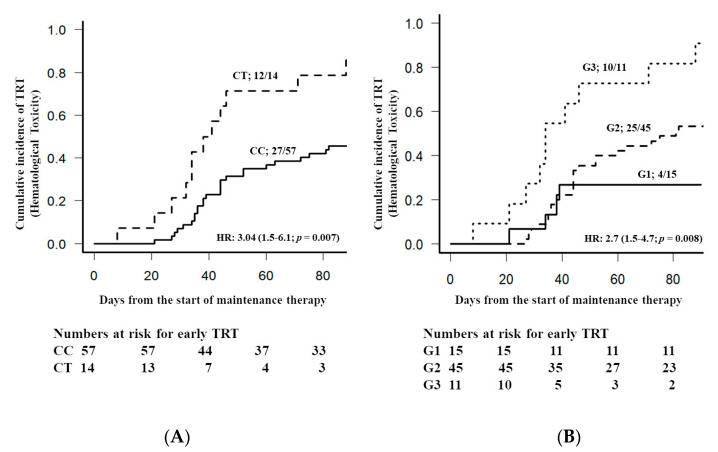
Association early hematological TRT during maintenance therapy in ALL patients with that of (**A**) *NUDT15* rs116855232(c.415C>T) variant in the whole observational cohort (*n* = 71) (**B**) *NUDT15* rs116855232(c.415C>T) variant and *FPGS* rs1544105 variant interaction (*n* = 73). G1 represents patients carrying no variant alleles for both loci, G2 represents patients carrying variant allele at either of these loci and G3 represents patients carrying variant alleles at both of these loci. The occurrence of TRT observed in non-carriers of *NUDT15* c.415C allele i.e.CC curve in plot “A” was partly explained by the variant allele carrier status for *FPGS* rs1544105, representing most of the G2 group in plot “B”. The group labels, number of events/total number of patients in each group, *p* values and hazards ratios for cumulative incidences of severe hematological toxicity are presented on the plots. Numbers at risk at each time point in each group indicates the number of patients without the event.

**Table 1 genes-11-00594-t001:** List of genes, genetic variants and their observed heterozygosity in the study cohort (*n* = 73).

Gene	rs ID of the SNP or Variant	Nucleotide Change	Effect on the Function /Location in Gene	Observed Heterozygosity Rate in the Cohort (%)
Variants in Genes Involved in 6-MP Pathway				
*NUDT15*	rs116855232 (*NUDT15*)	c.415C>T (NM_018283.3)	Decreased stability of the protein/exonic	0.197 (*n* = 71)
*TPMT*	rs1142345 (*TPMT**3C)	c.719A>G (NM_001346817.1)	Decreased efficiency and thermolabile/exonic	0.027 (*n* = 71)
Variants in Genes Involved in Folate/MTX Pathway				
a. DNA synthesis and methylation				
*DHFR*	rs1650694	g.3375C>T/G/A (NG_023304.1)	Upstream	0.90 (CT + CG) CA not detected
*MTHFR*	rs1801133	c.788C>T (NM_001330358.1)	Decreased function and thermolabile/exonic	0.23
	rs1801131	c.1409A>C (NM_001330358.1)	Decreased function and thermolabile/exonic	0.47
b. MTX polyglutamation				
*FPGS*	rs10106	g.15922T>C (NG_023245.1)	Unknown/3′UTR variant	0.55
	rs1544105	g.2572C>T (NG_023245.1)	Decreased function/Intron	0.55
*GGH*	rs3758149	g.4883C>T (NG_028126.1)	Increased function/5’UTR variant	0.48
	rs11545078	c.452C>T (NM_003878.2)	Unknown significance/exonic	0.25
c. MTX transporters				
*SLC19A1/* *RFC1*	rs1051266	c.80A>G (NM_001205206.1)	Unknown significance/exonic missense variant	0.79
*SLC01B1*	rs4149056	c.521T>C (NM_006446.4)	Decreased function/Exonic	0.07
*MDR or ABCB1*	rs1045642	c.3435T>C (NM_001348944.1)	Effects co-translational folding in nearby amino acids/exonic	0.40
	rs1128503	c.1236T>C (NM_001348944.1)	Unknown significance/exonic	0.36
	rs2032582	c.2677T>A/G (NM_001348944.1)	Decreased function/exonic	0.37 (TA + TG)

*DHFR-*Dihydrofolate Reductase, *FPGS-*Folylpolyglutamate Synthase, *GGH-*Gamma-glutamyl hydrolase, *MDR1* or *ABCB1-* Multi-drug resistance-1 gene, 6-MP-6-mercaptopurine, *MTHFR-*Methylenetetrahydrofolate reductase, MTX-Methotrexate, NUD15-Nucleoside diphosphate–linked moiety X-type motif 15, N is total number of patients, RFC1-Reduced folate carrier or solute carrier family 19 member 1 (SLC19A1), *SLCO1B1*-solute carrier organic anion transporter family member 1B1, TPMT-Thiopurine methyl transferase, UTR-untranslated region. Variants with unknown function are selected based on the previous literature on genetic associations especially with the methotrexate efficacy and toxicity.

**Table 2 genes-11-00594-t002:** Demographic and genotypic characteristics of patients with ALL according to early hematological toxicity (*n* = 73).

Variables	Severe (Grade 3–4) *n* (%)	Mild (Grade 0–2) or No Toxicity *n* (%)	OR	*p*-Value
**Age**				
0–18 (Mean ± SD)	9.09 ± 11.56	9.33 ± 11.57	NA	0.92
>18 (Mean ± SD)	27.50 ± 11.91	32.58 ± 12.04	NA	0.07
**Sex**				
Males	27 (67.5)	18 (55)	reference	
Females	13 (32.5)	15 (45)	0.58 (0.20–1.66)	0.33
**WBC (cells/mm^3^)**				
≤50,000	27 (67.5)	26 (79)	reference	
>50,000	13 (32.5)	7 (21)	1.77 (0.55–6.14)	0.31
**D8 blasts**				
Present	4 (10)	5 (15)	Reference	
Absent	36 (90)	28 (85)	1.59 (0.31–8.83)	0.72
**D15 blasts**				
Present	2 (5)	2 (6)	Reference	
Absent	38 (95)	31 (94)	1.22 (0.08–17.76)	1.00
**Hb (g%)**				
<11	34 (85)	23 (70)	Reference	
≥11	6 (15)	10 (30)	0.41 (0.10–1.45)	0.15
**Platelets (cells/mm^3^)**				
<100,000	32 (80)	19 (58)	Reference	
≥100,000	8 (20)	14 (42)	0.34 (0.10–1.07)	0.04 *
**Immunophenotype**				
Pre B or B-cell	23 (57.5)	17 (52)	Reference	
Pre T or T-cell	13 (32.5)	10 (30)	0.96 (0.30–3.09)	1.00
unknown	4 (10)	6 (18)		
**6-MP Dose(mg/m^2^, Mean ± SD)**				
MCP-841 protocol	3657 ± 1697	4100 ± 1973	NA	0.32
G-MALL protocol	6900 ± 260	6255 ± 2465	NA	0.66
**Dose MTX (mg/m^2^, Mean ± SD)**				
MCP-84136 22	110.54 ± 44	134 ± 49	NA	0.07
G-MALL3 8	1326 ± 1762	378 ± 39	NA	0.44
**6-MP Dose reduction (*n* = 68)**				
No	27 (71)	29 (97)	reference	
Yes	11 (29)	1 (3)	11.47 (1.48–524.29)	0.008 *
**MTX dose reduction (*n* = 69)**				
No	32 (82)	29 (97)	reference	
Yes	7 (18)	1 (3)	6.21 (0.73–295.1)	0.13
***NUDT15* rs116855232 (*n* = 71)**				
CC	27 (67.5)	30 (91)	reference	
CT	12 (30)	2 (6)	6.51 (1.27–65.16)	0.01 *
***TPMT* rs1142345 *3C (*n* = 71)**				
AA	38 (95)	31 (94)	reference	
AG	1 (2.5)	1 (3)	0.81 (0.01–66.04)	1.00
***MTHFR* rs1801133**				
CC	30 (75)	25 (76)	reference	
CT	10 (25)	7 (21)	1.04 (0.31–3.54)	1.00
TT	0	1 (3)		
***MTHFR* rs1801131**				
AA	16 (40)	13 (39)	reference	
AC	18 (45)	16 (48)	0.97 (0.34–2.76)	1.00
CC	6 (15)	4 (12)		
***RFC1 /SLC19A1* rs1051266**				
AA	6 (15)	9 (27)	reference	
GA	34 (85)	24 (73)	2.10 (0.58–8.21)	0.25
***SLCO1B1* rs** **4149056**				
TT	37 (92.5)	31 (93.9)	reference	
TC	3 (7.5)	2 (6.1)	1.25 (0.13–15.89)	1.00
***GGH*** **rs3758149**				
CC	20 (50)	12 (36.4)	reference	
CT	18 (45)	17 (51.5)	0.57 (0.19–1.62)	0.34
TT	2 (5)	4 (12.1)		
***GGH* rs11545078**				
CC	31 (77.5)	23 (69.7)	reference	
CT	9 (22.5)	9 (27.3)	0.67 (0.20–2.17)	0.59
TT	0 (0)	1 (3)		
***MDR1* rs1045642**				
CC	6 (15)	5 (15.2)	0.81 (0.29–2.27)	0.81
CT	15 (37.5)	14 (42.4)	reference	
TT	19 (47.5)	14 (42.4)		
***MDR1* rs1128503**				
CC	4 (10)	6 (18.2)	0.43 (0.15–1.22)	0.10
CT	12 (30)	14 (42.4)	reference	
TT	24 (60)	13 (39.4)		
***MDR1* rs2032582**				
GG	2 (5)	5 (15)	0.43(0.15–1.22)	0.10
GA	1 (2)	1 (3)	reference	
GT	9 (22)	12 (36)		
TA	4 (10)	2 (6)		
TT	24 (60)	13 (39)		
***DHFR* rs1650694**				
CC	0 (0)	2 (6)	reference	
CG	30 (75)	25 (76)	Inf (0.22–Inf)	0.20
CT	7 (18)	4 (12)		
GT	3 (8)	2 (6)		
***FPGS* rs10106**				
TT	6 (15)	11 (33.3)	reference	
TC	25 (62.5)	15 (45.5)	2.79 (0.80–10.62)	0.09
CC	9 (22.5)	7 (21.2)		
***FPGS* rs1544105**				
CC	6 (15.4)	12 (36.4)	reference	
CT	25 (64.1)	15 (45.5)	3.18 (0.93–12.01)	0.05
TT	9 (20.5)	6 (18.2)		

CI: confidence interval, *DHFR:* Dihydrofolate Reductase, *FPGS:* Folylpolyglutamate Synthase, *GGH:* Gamma-glutamyl hydrolase, Inf: infinite; *MDR1* or *ABCB1:* Multi-drug resistance-1 gene, 6-MP:6-mercaptopurine, *MTHFR:* Methylenetetrahydrofolate reductase, MTX: Methotrexate, NA:-not applicable; NUDT15: Nucleoside diphosphate–linked moiety X-type motif 15, N is total number of patients, OR: odds ratio; RFC1: Reduced folate carrier or solute carrier family 19 member 1 (SLC19A1), *SLCO1B1*: solute carrier organic anion transporter family member 1B1, TPMT: Thiopurine methyl transferase, WBC and platelet count at the time of diagnosis. 6-MP and MTX dose reduction also includes data that is consequence of the occurrence of toxicity. The data is presented as *n* (%) and ‘n’ is the number of subjects in that particular group. ref-reference, nonparametric test was used for continuous variables and Fisher’s exact was used for categorical variables to obtain the odds ratio (odds of variant/odds of reference or normal) in a dominant mode of inheritance. ** *p* value (two-sided, corrected for multiple testing) < 0.05 is considered as significant. * *p* values (two sided, uncorrected for multiple testing) are also mentioned in the manuscript.3.1. Association of Clinical Risk Factors and Genetic Variants with Early TRT.

**Table 3 genes-11-00594-t003:** Multiple testing *p* values obtained using Benjamini–Hochberg method.

Gene	Variants Tested	Original *p* Value Obtained for Cumulative Incidence of TRT	Benjamini-Hochberg Adjusted *p* Value
*NUDT15 & FPGS*	*NUDT15*c.415C>T (rs116855232) & *FPGS* (rs1544105) interaction	0.00057	0.008 **
*NUDT15*	*NUDT15*c.415C>T (rs116855232)	0.001005	0.007 **
*FPGS*	rs1544105	0.08675216	0.325
	rs10106	0.1419533	0.354
*GGH*	rs3758149	0.09727433	0.291
	rs11545078	0.4538174	0.756
*RFC1*	rs1051266	0.1912443	0.409
*MDR1*	rs1045642	0.3542364	0.664
	rs2032582	0.5319389	0.797
	rs1128503	0.02607358	0.131
*MTHFR*	rs1801133	0.6694676	0.912
	rs1801131	0.8705498	0.870
*DHFR*	rs1650694	0.6743685	0.842
*TPMT*	rs1142345	0.7680374	0.886
*SLCO1B1*	rs4149056	0.7892506	0.845

** Significant (two-sided, corrected for multiple testing) using an FDR of 0.05, after multiple testing correction using Benjamini-Hochberg method.

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
