# Peer review of "Association of NUDT15 c.415C>T and FPGS 2572C>T Variants with the Risk of Early Hematologic Toxicity During 6-MP and Low-Dose Methotrexate-Based Maintenance Therapy in Indian Patients with Acute Lymphoblastic Leukemia"

_genes, 2020, doi:10.3390/genes11060594_

Round 1
Reviewer 1 Report
The authors investigated the association of genetic polymorphism with the risk of early hematological toxicity during low-dose methotrexate and 6-mercaptopurine-based maintenance therapy in Indian acute lymphoblastic leukemia patients, and suggested NUDT15*3 and FPGS 2527G>A variants as the potential risk stratifying markers. Although the manuscript describes the details of the study, there are some aspects that should be addressed prior to being considered for publication.
Major comments
- It seems to be a conflict between the following two points. Although authors discussed about the contradictory effect of FPGS variant, it is recommended to discuss hypotheses for this phenomenon in more detail.
- FPGS converts MTX to MTXPG, which inhibits xanthine oxidase, and consequently enhances the risk of developing 6-MP-induced myelosuppression.
- In this study, carriers of FPGS variant related to decreased protein function showed an increased trend towards the risk of early TRT.
- Page 14, line 384-386: In conclusions, authors stated that “Implementation of preemptive NUDT15, FPGS, and TPMT genotyping to guide 6-MP dosing is worthy of prospective evaluation in limited resources settings, especially in terms of health economic evaluation.” However, TPMT variant (rs1142345) showed no significant association with the risk of early TRT in this study, so I wonder why authors chose TPMT as a preemptive genotyping marker in addition to NUDT15 and FPGS, which had significant association with the risk of early TRT. Even the effect of TPMT mutation is already well known, but it is out of scope of this study.
- Page 3, line 91-93: Authors evaluated 14 candidate variants from 9 key genes that affect the PK and PD of MTX and 6-MP. It is recommended to explain in detail about why the authors chose these variants and genes for the investigation, and describe additionally about which pathway the corresponding genes are involved in.
- Page 10, line 243~251: The MTXPG levels were not associated with the toxicities. However, FPGS variation, one of important finding of this study, can affect the conversion of MTX to MTXPG, and then change the MTXPG level. The two findings are controversial. Please discuss these finding in detail.
- Table 2: It seems to need some revision especially in ‘OR’ column as follows.
- Specify the reference group of each demographic characteristics such as age, sex and WBC, etc, similar to genetic variation.
- It is confusing that some values of OR were written in the line of reference group, and others in the line of test group. So, unify the position of the line where values of OR are written.
- In genotypic characteristics, OR for NUDT15 was calculated as ‘odd of variant/odd of reference’, but ORs for other genes seem to be calculated as ‘odd of reference/odd of variant’. Please re-calculate ORs with the corrected calculation formula.
- Throughout the manuscript (including Table 2 and Table 3), ‘rs1544104’ and ‘rs1544105’ are mixed as an expression of FPGS The expression for the FPGS variant should be corrected and unified.
Minor comments
- Page 5, line 156-157: The sentence “As expected, many hypotheses failed possibly represent false negatives due to low sample size.” is required for proof-reading for grammar.
- Page 9, line 206-207: It seems that rounding of the values of cumulative incidence of TRT for G1 (26.6%) and G2 (55.5%) is wrong. Re-calculate the values in the revised manuscript.
- Page 12, line 303: Authors described the heterozygosity of FPGS variant as ‘48%’, but the corresponding value in Table 1 is ‘0.56’. One of the values should be corrected.
- Page 13, line 343: The term ‘clearance volumes’ seems pharmacokinetically inappropriate.
- Table 1: It seems easier to understand the study results for the readers if authors describe more intuitionally about ‘Effect on the function/Location’ (ex. ‘Decreased function’ or ‘Increased function’).
- Table 1: The term ‘heterozygosity’ should be revised to more appropriate term such as ‘allele frequency’ or ‘heterozygosity rate’.
- Table 3: Please add the corresponding gene name for each variant in ‘Variants tested’ column.
Author Response
We thank the reviewer for critical remarks to improve the presentation of the data and observations in the manuscript.
Please find attached the responses to the reviewer comments point-wise. Responses are in the red color font.
The corrections and modifications in the revised manuscript file are also highlighted in red color font.
Thank you for giving us the opportunity to improve the manuscript.

Reviewer 2 Report
The authors study a cohort of 73 ALL patients of South Asian (Indian ancestry). They perform clinical assessments and primarily monitor treatment related toxicities within the first 100 days of maintenance therapy that featured 6-MP and methotrexate treatments. In this study, blood samples were tested for methotrexate metabolite levels and to isolate genomic material. The latter samples were genotyped for 14 variants in genes associated with drug metabolism, transport and toxicity. Association studies identified that variants in NUDT15 were associated with early onset TRT and this could be improved by including FPGS variants in the analyses. The authors also considered relapse and dose reductions in their study.
This is a well-written manuscript with fairly well-justified findings.
The authors detail possible interactions between 6-MP and MTX including reference to Fig. S1. They authors state that MTX inhibits XO (providing only a single reference that does not include a direct citation). This does not appear to be a well-documented drug-gene interaction. MTX inhibit folate synthesis enzymes and XO is inhibited by allopurinol. The authors should include additional citations for their argument. If the argument is well-supported (this is not yet clear) then Fig. S1 can be modified to show that MTX inhibits XO, which it currently does not do.
The assertion that combining genotypes for NUDT15 and FPGS offers improved sensitivity is not clearly made by Fig. 1B. Can the authors compare the CT curve from panel A with the G3 curve from panel B on the same graph so that we can directly examine the effect of adding FPGS into the model? Furthermore, the sensitivity and specificity values mentioned in the discussion (lines 307-309) are not explicitly stated in the results and this reviewer is unclear how these values were derived because they do not appear to come from Table 3 as stated. The authors should consider including a better description of what the “Numbers at risk for early TRT” actually reflect. Judging by their alignment they correspond to days post-treatment but the figure legend and results section do not provide much information about the utility of these numbers.
Minor comments:
Line 231 . revise to indicate that ABCB1 is also known as MDR1.
Fig. S3 – the “Numbers at risk for early TRT” are the same for the left and right panel - is this accurate?
Fig. S4 – the “Numbers at risk for” are the same for panels A and B – is this accurate?
Author Response

(The authors gave the same response as above.)

Round 2
Reviewer 1 Report
The revised manuscript is more clear than previous one, but there are still some points that need to be corrected.
1.Table 2: There are calculation errors in ‘OR’ column, which are in discord with the calculating formula the author had added in the revised manuscript (odds of variant/odds of reference or normal). The followings are only examples, so please carefully check and correct the ‘OR’ column.
A. Especially, in demographic characteristics, the author calculated the ‘OR’ as ‘odds of reference/odds of variant’.
B. In case of NUDT215 rs116855232, the correct OR value seems to be ‘6.67’, and in case of GGH rs3758149, the corresponding value seems to be ‘0.64’. Other values should be confirmed.
2.The author stated that “The cumulative incidences represented for G1 group had 15 patients with 4 people having events among them constituting 26.6%; whereas Group 2 had 20 events per 45 patients constituting 44.44 %, and for G3 it was 10 events per 11 individuals constituting 90.9 %,” in the response for comments. However, in figure 1B, the number of G2 events is still described as ‘25’, so revised it to the correct value. Also, the cumulative incidence for G1 seems to be ’26.7%’.
Author Response
we thank the reviewer for precision in the remarks and suggestions to improve the presentation of the data (in a uniform way) in the manuscript.
Please find attached the document with responses to the reviewer comments. The modifications in the manuscript are highlighted in blue color font.
